# Integrating evolutionary dynamics into treatment of metastatic castrate-resistant prostate cancer

Jingsong Zhang[1], Jessica J. Cunningham[2], Joel S. Brown[2,3] & Robert A. Gatenby[2,4]

Abiraterone treats metastatic castrate-resistant prostate cancer by inhibiting CYP17A, an enzyme for testosterone auto-production. With standard dosing, evolution of resistance with treatment failure (radiographic progression) occurs at a median of ~16.5 months. We hypothesize time to progression (TTP) could be increased by integrating evolutionary dynamics into therapy. We developed an evolutionary game theory model using Lotka–Volterra equations with three competing cancer "species": androgen dependent, androgen producing, and androgen independent. Simulations with standard abiraterone dosing demonstrate strong selection for androgen-independent cells and rapid treatment failure. Adaptive therapy, using patient-specific tumor dynamics to inform on/off treatment cycles, suppresses proliferation of androgen-independent cells and lowers cumulative drug dose. In a pilot clinical trial, 10 of 11 patients maintained stable oscillations of tumor burdens; median TTP is at least 27 months with reduced cumulative drug use of 47% of standard dosing. The outcomes show significant improvement over published studies and a contemporaneous population.

[1] Department of Genitourinary Oncology, Moffitt Cancer Center & Research Institute, Tampa, FL 33612, USA. [2] Department of Integrated Mathematical Oncology, Moffitt Cancer Center & Research Institute, Tampa, FL 33612, USA. [3] Department of Biological Sciences, University of Illinois at Chicago, Chicago, IL 60607, USA. [4] Department of Diagnostic Imaging and Interventional Radiology, Moffitt Cancer Center & Research Institute, Tampa, FL 33612, USA. Correspondence and requests for materials should be addressed to R.A.G. (email: Robert.Gatenby@moffitt.org)

Evolution of resistance is a common cause of cancer treatment failure and tumor progression but explicit incorporation of intratumoral Darwinian dynamics in therapeutic trials is rare[1]. In fact, the conventional treatment strategy, which administers cytotoxic drugs at maximum tolerated dose (MTD) until progression, can be evolutionarily unwise because it strongly selects for resistant phenotypes and eliminates potential competitors. These Darwinian dynamics—termed "competitive release"[2]—can lead to rapid proliferation of resistant populations.

A number of recently developed treatment strategies have applied evolutionary principles to prolong tumor control by inhibiting the emergence of treatment-resistant populations[3–6]. These strategies[7] typically exploit the evolutionary costs of synthesis, maintenance, and operation of the molecular machinery needed to survive treatment. The benefits of resistance exceed costs during therapy. However, in the absence of treatment, particularly in the resource-limited tumor microenvironment, this cost renders resistant cells less fit than sensitive phenotypes[8]. Thus, appropriately timed withdrawal of treatment can allow residual populations of sensitive cells to exploit their fitness advantage at the expense of the less-fit resistant phenotypes. While discontinuation of treatment allows tumor regrowth, the resistant subpopulation remains small so that retreatment with the same drug(s) remains effective (Fig. 1).

Evolution-based treatment strategies have successfully controlled breast and ovarian cancers, often indefinitely, in pre-clinical experiments[2, 3, 9]. However, translation to a clinical setting has remained elusive.

Conceptually, therapy and the evolution of resistance represent an evolutionary game between the cancer and oncologist (not unlike a predator-prey game) and between the different cancer cell types[10, 11]. As a game, the cancer cells are the players, their heritable phenotypes their strategies, and payoffs take the form of proliferation and survival. A cancer cell's survival and proliferation can be influenced by its own phenotype, the phenotypes of others and the overall abundances of the different cancer cell types[12, 13]. Mathematical models provide valuable tools for formulating hypotheses and for evaluating different scenarios pertaining to the interactions between cancer cell types and therapy[14–16].

Here, we focus on the treatment of metastatic prostate cancer (mPC). First-line treatment of mPC uses androgen deprivation therapy (ADT), but nearly all men progress to a metastatic castrate-resistant prostate cancer (mCRPC) stage. A common mechanism of resistance to ADT is increased expression of CYP17A1[17, 18], a key enzyme for androgen synthesis[19, 20]. This generates an autocrine loop that replenishes intratumoral testosterone concentrations. Abiraterone acetate, a CYP17A1 inhibitor, reduces PSA, and improves overall survival. In subjects who

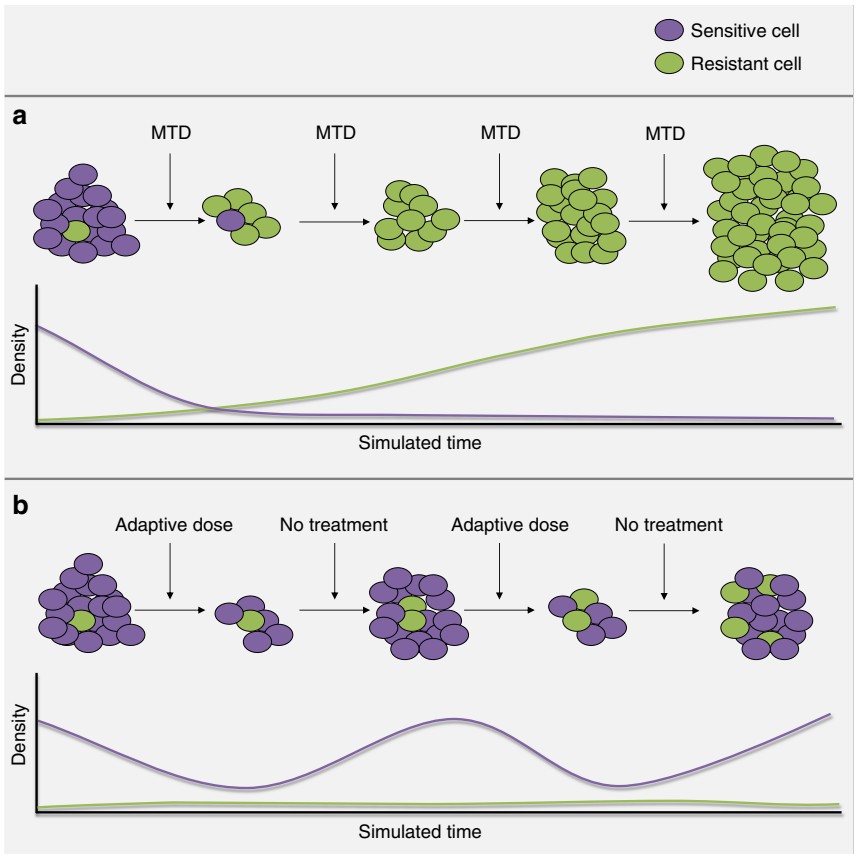

**Fig. 1** Illustration of the designed evolutionary dynamics in adaptive therapy. **a,b** The purple cells are sensitive to the treatment and the green cells are resistant. The graphs represent the simulated density of each population over time during treatment. The top row represents standard therapy in which the maximum tolerated dose is given continuously after initiation. The cells sensitive to treatment are eliminated quickly. This intensely selects for cells that are resistant to the treatment, in this case T− cells, and eliminates the competition effects of the T+ population, resulting in competitive release with rapid treatment failure and tumor progression. The bottom row represents an evolution-based strategy in which therapy is halted before all of the sensitive cells are eliminated. In the absence of therapy, the sensitive cells out-compete the resistant cells due to their fitness advantage. This "steers" the tumor back to the pretreatment so that it remains sensitive to treatment. The resistant cells, or T− population, will increase slightly with each cycle so that this treatment eventually fails. However, mathematical models demonstrate control may be durably maintained for up to 20 cycles - significantly longer than continuous therapy.

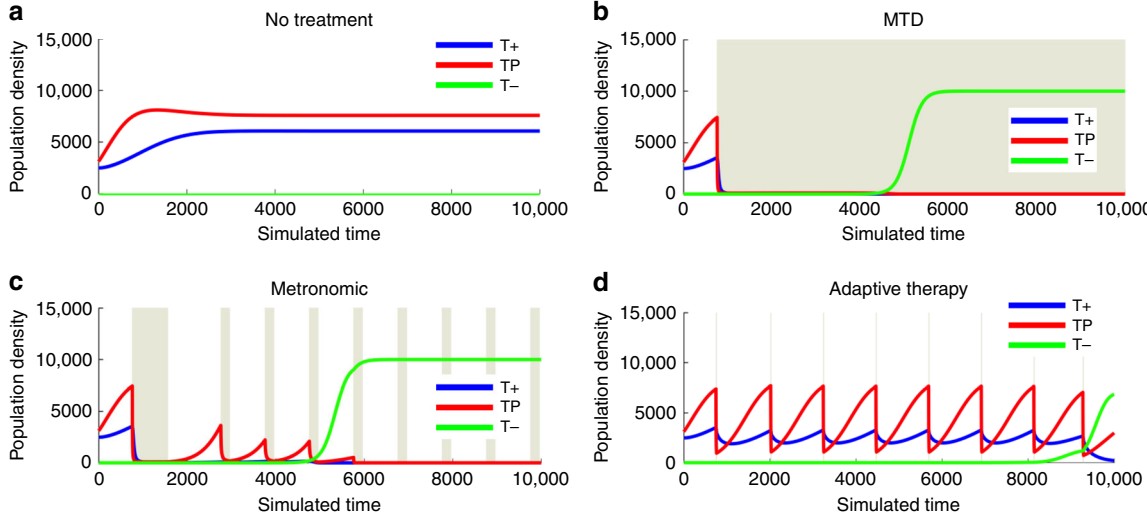

**Fig. 2** Simulation results. Computer simulations of mCRPC growth during conventional maximum tolerated dose, metronomic, and adaptive application of abiraterone where the gray background indicates administration of abiraterone in Patient #1. **a** shows underlying population dynamics of a tumor if left untreated. **b** shows continuous application of abiraterone resulting in competitive release of T− cells and tumor progression. **c** panel shows a metronomic therapy similar to ADT intermittent therapy study where the lengthy induction period and further abiraterone is given at predetermined intervals. This shows that the benefit gained from adaptive therapies is not just the decrease in drug dosage but is indeed the evolutionary guided timing of the cycles. **d** shows a short administration of abiraterone decreasing the TP and T+ cells. However, abiraterone is discontinued when the PSA falls below 50% of the pretreatment value (Fig. 3). This permits recovery of TP and T+ cells, reverses the increase in T− cells, and prevents competitive release. After each treatment cycle, the tumor subpopulations remain nearly identical in size and composition

initially respond to abiraterone, median time to PSA progression ranges from 5.8 to 11.1 months, depending on the study and if docetaxel was previously administered[20–23], and median time to radiographic progression is about 16.5 months.

A previous trial (SWOG 9346) investigated intermittent ADT with goals of improving quality of life and delaying onset of hormone resistance[24, 25]. Patients with newly diagnosed castration sensitive mPC were randomized into continuous or intermittent ADT treatment. After a median follow up of 9.8 years, neither regimen proved superior. While this trial attempted to use evolutionary dynamics to improve treatment outcomes, our analysis found it lacked an explicit link between the treatment schedule and intratumoral evolutionary dynamics. For example, the protocol[25] started with an 8-month "induction period" with continuous high-dose therapy. Treatment was discontinued only if PSA < 4 ng/ml. In our computational model (see below), we found this trial design, by eliminating nearly all of the sensitive cells during the induction period, promoted competitive release virtually identical to continuous high-dose therapy.

We suggest that the successful application of evolutionary principles to the clinical treatment of prostate cancer requires formal mathematical models that frame the complex, often non-linear, interactions that define response and resistance to therapy in individual patients. Here we present a methodology for integrating intratumoral Darwinian dynamics in the clinical treatment of mCRPC. We first define relevant intratumoral subpopulations based on their interactions with the critical treatment factor—testosterone. Observations of clinical specimens[17] reveal three competing phenotypes: (i) T+ cells requiring exogenous androgen; (ii) TP (testosterone producing) cells expressing CYP17A1 and producing testosterone; and (iii) T− cells that are androgen-independent and resistant to abiraterone. We frame the population dynamics before and during abiraterone therapy using a game theoretic model built on evolutionary first principles with parameter estimates derived from clinically available data.

Computer simulations of conventional MTD treatment were consistent with observed clinical outcomes with a rapid increase

of resistant (T−) populations. The modeling results then demonstrated how an adaptive therapy treatment strategy synchronized with intratumoral evolutionary dynamics may prolong time to progression and substantially decrease total drug dose. The evolution-based strategy suggested by the model was then applied in a pilot clinical trial.

Here we present the mathematical and computational methods used to integrate evolutionary dynamics into a clinical trial, and the results of an interim analysis of the trial cohort.

## Results

**Model simulations**. Model simulations (Fig. 2) demonstrate that an untreated tumor will result in a large tumor volume as various mixes of TP, T+, and T− cells grow to lethal population sizes. First-line androgen deprivation therapy (ADT) targets the T+ population but not the TP or T− phenotypes. However, in this adaptive landscape, the T+ cells can persist as "cheaters" in the sense that they "freeload" on the testosterone produced by TP cells while contributing nothing to the cost of this production. In mCRPC, conventional, continuous application of abiraterone at MTD eliminates TP cells and T+ cells, resulting in competitive release of T− cells and tumor progression. Simulations of intermittent therapy following a long "induction period" of MTD treatment, as reported in a prior trial[25], showed progression equivalent to continuous MTD treatment.

Model simulations demonstrate evolutionary dynamics can be exploited to preserve therapy-sensitive TP and T+ cells. One such strategy withdraws abiraterone when the PSA drops to half of its pretreatment value (Fig. 3). This permits the tumor to regrow but, in the absence of therapy, the sensitive cells are fitter and thus remain the dominant population. In this way, the tumor population after treatment is nearly identical to the pretreatment tumor allowing retreatment with abiraterone to maintain tumor control over multiple cycles.

The outcomes from adaptive therapy are patient-specific and depend on the composition of the tumor prior to abiraterone

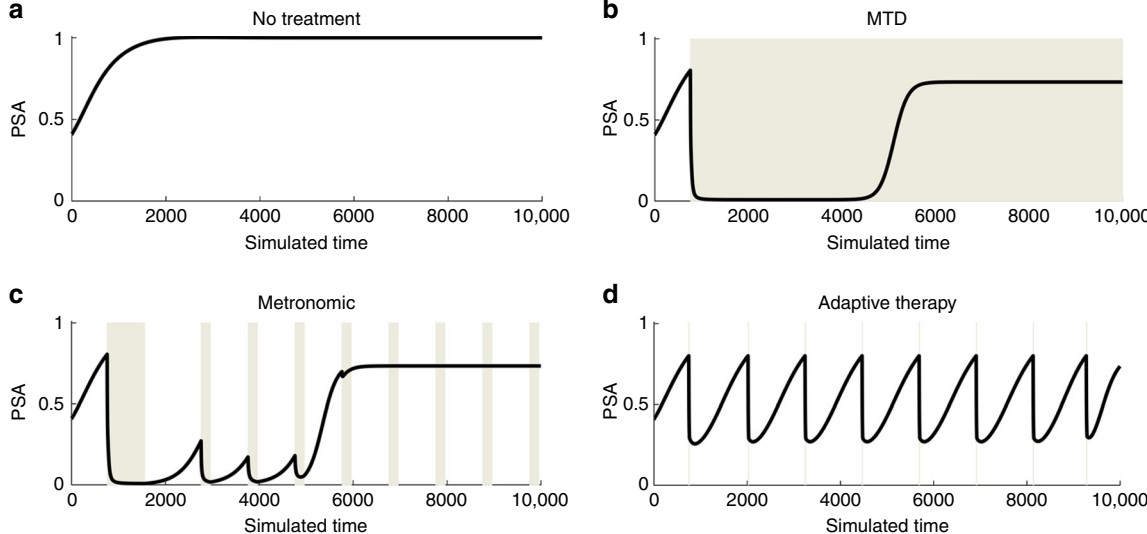

**Fig. 3** Computer simulations of PSA under varying treatment conditions. The gray background indicates administration of abiraterone in Patient #1. **a** demonstrates the PSA dynamics if no treatment was administered. **b** shows the classic PSA dynamics of maximal tolerated dose MTD, where a large response is maintained until PSA progression. **c** show the PSA dynamics for a metronomic therapy where treatment is not synchronized to patient-specific PSA dynamics. **d** shows the PSA dynamics of the clinical trial protocol, where PSA decreases to 50% of the baseline PSA and is allowed to return back to baseline before another dose of abiraterone is given

therapy (Fig. 4). A small population of T+ cells results in longer cycle times as the PSA value takes longer to reach the retreatment PSA value (see, for example, patients 1003, 1005, and 1007 in Fig. 5). Durable control of T− cells during the PSA trough of abiraterone therapy is largely provided by the TP cell population. Alternatively, a higher density of T+ cells results in shorter cycle times. Both scenarios allow a small increase of the T− cells during the PSA trough of abiraterone therapy. This results in a slow, cycle to cycle increase in the population of resistant cells that eventually leads to treatment failure, though after a significantly longer time to progression than MTD.

Varying the relative values of carrying capacities among cancer cell types results in small and insubstantial changes in the overall results (Supplementary Table 1). Increasing the magnitude or spread of the inter-type competition coefficients decreases the likelihood that all three cell types coexist within the pre-abiraterone tumor, and increases the spread in cell type frequencies from rarest to most common (Supplementary Table 2).

The times to competitive release, which corresponds to clinical progression, for the simulated therapy regimens are shown in Table 1. Here we see that adaptive therapies provide equivalent or increased time to progression under any initial tumor condition (Supplementary Table 3).

**Clinical trial and statistics.** At the time of writing, 11 patients (Table 2) have been on the trial sufficiently long (>10 months) to be compared with a contemporaneous cohort and historic controls. As predicted by the mathematical models, cycles of PSA were observed (Fig. 4) with cycle lengths ranging from 3 months to >1 year. Four patients have completed two adaptive cycles and all four patients achieved > 50% decline of PSA when abiraterone was restarted at cycle 3 (primary end point). The current clinical status of the trial cohort is shown in Fig. 5. One patient developed PSA and radiographic progression at month 11. Two patients have exhibited PSA progression at 21 and 28 months but remain on trial with no radiographic progression. Two patients declined to continue GnRH suppression with Leuprolide (an ADT that is continuously administered as part of any abiraterone therapy) at

months 17 and 20, respectively, but continue taking abiraterone. None of the 11 patients have experienced grade 2 or above adverse events that would necessitate stopping abiraterone and prednisone.

Also notable is the reduction in cumulative drug dose. Because of the on/off cycling of abiraterone, the adaptive therapy cohort has received an average cumulative dose of just 47% compared to continuous dosing used in standard of care (Fig. 5; Table 2). Three of the 11 patients have received < 25% of the cumulative SOC dose.

A study by Ryan et al.[22] provides an historic group for comparing time to PSA progression and radiographic progression while on continuous abiraterone therapy. In the phase 3 AA 302 study, they report median times of 11.1 and 16.5 months for PSA progression and radiographic progression, respectively. Our adaptive therapy cohort has not reached a median time to either PSA or radiographic progression but, at the time of submission, they can be no less than 27 months.

PSA progression has been observed in three patients at 11, 21, and 28 months. Given the 546 patients of the Phase 3 trial, we can use a contingency table to test for efficacy of the adaptive therapy. Since just one of the 11 patients can have a time to PSA progression of <11.1 months, our trial group must have a significantly higher median (Fisher exact test $p = 0.006$).

In the phase 3 trial, the reported median of time to radiographic progressions (TTP) was 16.5 months. Although the mean TTP is not reported, it cannot be higher than 17.2 months (true for either a Poisson or Binomial distribution of progression times). If we assume that all of our patients suddenly progress radiographically at time of writing (only one actually has), the mean TTP in the treated group cannot be shorter than 24 months. The Poisson exact test comparing the 11 patients to a mean of 17.2 gives $p < 0.001$ ($z = 4.10$). Using the Poisson exact test to test for the likelihood that the adaptive therapy has a mean of <17.2 gives $p = 0.98$. Even with this small sample size, for radiographic progression, we can demonstrate superiority of adaptive therapy.

A contemporaneous (treated within the past 5 years by the same team of physicians) cohort was generated from 16 patients

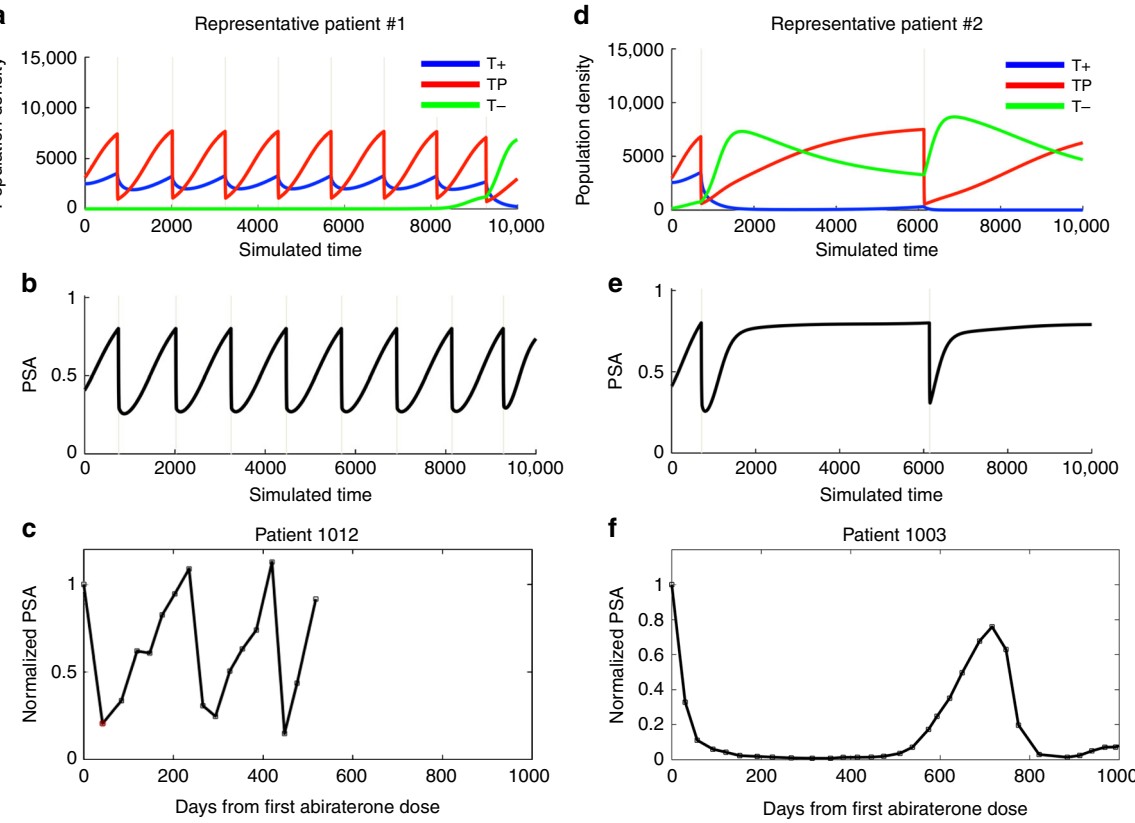

**Fig. 4** **a,b,d**, and **e** show computer simulations demonstrating variation in cycle length. Time between treatments is shown to vary based on the competition coefficients of the matrix and the resulting prevalance of T+ cells. Panels **a** and **b** show Patient #1's fast cycling dynamics as the large T+ population contributes to the PSA reaching the treatment PSA level quickly. The durable control of T− cells is provided by the TP cell population. Alternately, panels **d** and **e** Patient #2 shows a low density of T+ cells resulting in long cycles. The variation in patient cycling rates explains the limitation of intermittent therapies administered without synchronization with underlyling evolutionary dynamics. Panels **c** and **f** show actual PSA fluctuations in two of the clinical trial patients with associated abiraterone administration. The first dose of abiraterone is given at day 0. In each case, the PSA is normalized to its value on day 0. The drug is withdrawn when the PSA falls below 50% of the original value. It is withheld until the PSA returns to its initial value (corresponding to the PSA peaks). This explicit incorporation of Darwinian principles into treatment allows drug administration to synchronise with patient-specific intratumoral evolutionary dynamics

with chemotherapy naive mCRPC who exhibited an initial PSA decline of >50% of baseline while on continuous abiraterone as standard of care. Identical to our study, these patients had ECOG 0–1 performance status (PFS), no visceral metastases and had follow ups for at least 6 months after their first dose of abiraterone. This cohort consists of patients who would have been eligible for the trial but were treated prior to its opening or elected not to participate. In this cohort, 14 of 16 (as opposed to 1 of 11 in our trial cohort) have shown both PSA and radiographic progression. The median times to PSA and radiographic progression are 9 and 14 months, respectively. A Fisher's exact test on the 2 × 2 contingency table (therapy group × radiographic progression status) is highly significant ($p < 0.001$). Note that this test is conservative since the control group has smaller mean times on therapy either to progression or still on therapy compared to the adaptive therapy patients. Additionally, we can compare time to radiographic progression under the conservative assumption that all patients had progressed. Performing a $t$-test (two sample, unequal variances) on time to radiographic progression, the 16 contemporaneous control (mean 13.2 months) and the 11 adaptive therapy patients (mean of 24) shows a significant difference ($t_{24} = 3.38$, $p < 0.005$).

Thus, the adaptive therapy results are consistent with the dynamics predicted by the mathematical model. Furthermore, even with just 11 patients we see statistically significant

improvement over the 16 comparable contemporaneous patients at Moffitt undergoing standard of care and over the 546 patients in an historic study of abiraterone standard of care.

## Discussion

Many effective therapies for disseminated, metastatic cancers are available but evolution of resistance almost inevitably leads to treatment failure and tumor progression. Despite the critical role of evolution for therapy outcomes, Darwinian dynamics are rarely integrated into clinical oncology protocols. Prevention of hormone resistance was an explicit goal of prior studies using intermittent ADT but the trial design did not explicitly define, quantify, or model the key Darwinian forces governing response and resistance to treatment[24, 25]. Thus, efforts to translate evolutionary dynamics into a clinical setting have generally used informal, non-quantitative approaches. In fact, when the trial design, which included an 8 month induction period using maximum dose, was simulated in our models, the evolutioanry dynamics of resistance were identical to those of the standard of care, continuous MTD treatment - the outcome that was actually observed in the trial.

Here we demonstrate a method for using mathematical models to integrate evolutionary principles into abiraterone therapy for mCRPC. Interim analysis of a pilot trial demonstrate that this

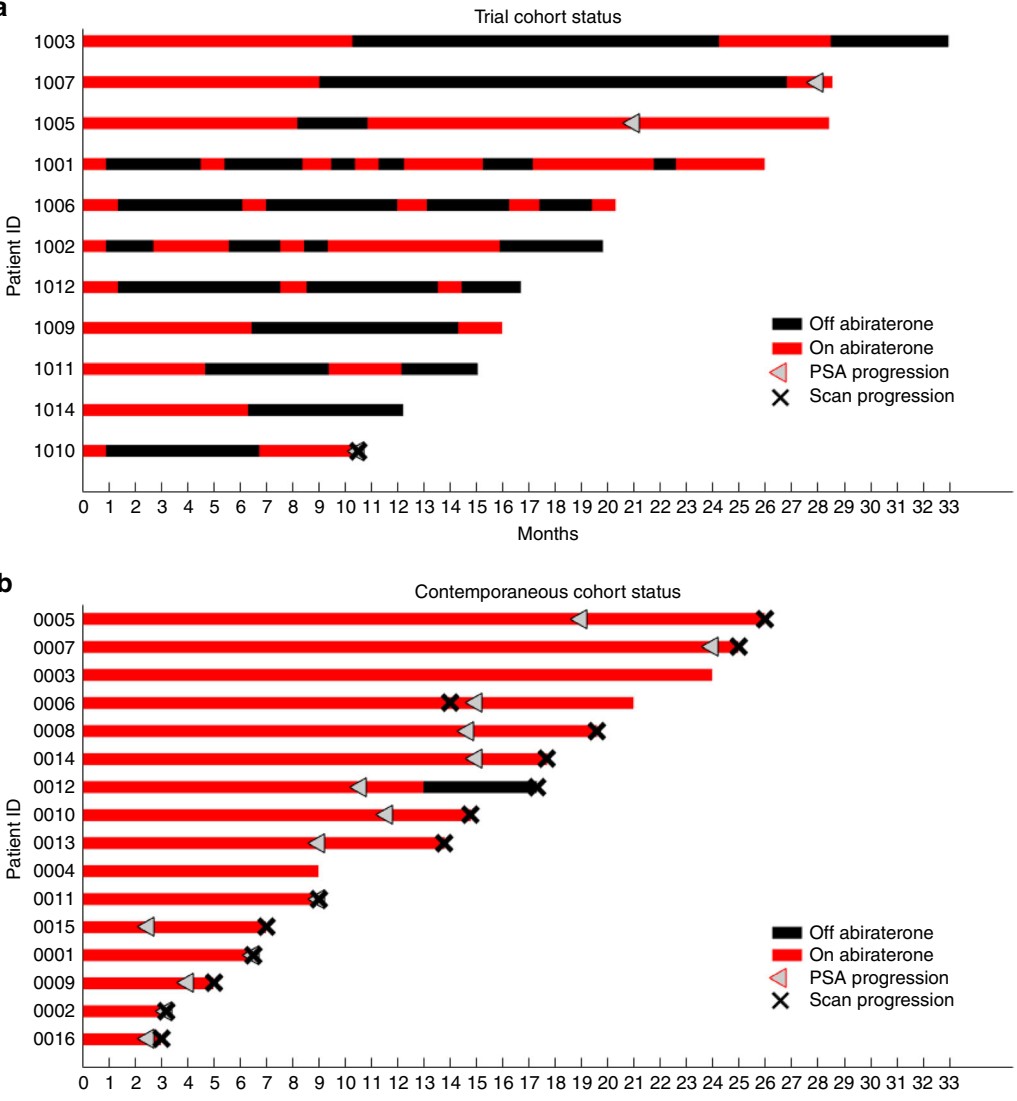

**Fig. 5** A summary of the status of the 11 patients in the pilot trial (**a**) and 16 patients in the contemporaneous cohort (**b**). In the SOC group, 14 of 16 have progressed radiographically compared to 1 of 11 in the adaptive therapy cohort. Cumulative dose of abiraterone in the adaptive therapy patients was 47% of SOC

approach is clinically feasible. Furthermore, the approach significantly increases time to progression while reducing the cumulative drug dose to <1/2 that of standard of care. While abiraterone generally has few side effects, the reduced dose does affect the monthly pharmacy cost, which may exceed $6000 for SOC dosing.

Estimating intratumoral evolutionary and ecological dynamics during therapy is challenging since, as shown above, relevant clincal data are often sparse. However, it should be possible to significantly refine mathematical models and further improve patient outcomes by incorporating additional diagnostic tools, such as circulating tumor cells[26, 27] and DNA[28, 29] as well as applying novel analytic methods to conventional clinical imaging[29].

While favorable, our results should be viewed with caution. The application of this strategy to mCRPC will require further study in larger clinical trials. We have already identified strategy modifications that will likely further improve outcomes. Furthermore, our model rests on the assumption that the key subpopulations compete with each other and, therefore, have some

degree of spatial mixing (Fig. 1). This appears to hold for mCRPC, but may not for other tumor types. Indeed, spatial mixing in mCRPC may be unusually high due to the relatively small tumor burdens and recent passage through an evolutionary bottleneck imposed by ADT. Spatially-explicit models with multiple drugs targeting different subpopulations may be necessary in larger, regionally-heterogeneous tumors.

We conclude that our initial results are encouraging. We advocate larger clinical trials of abiraterone therapy using evolutionary first principles and game theoretic mathematical models.

## Methods

**Evolutionary subpopulations**. We view each prostate cancer metastases as an open complex dynamic system. It is complex because it has many components, dynamic because those components interact with each other in complicated, often non-linear ways and open because it extensively interacts with the host[30]. A mathematical model that attempts to capture all of these dynamics would be intractable and, in any case, not useful. It could not be parameterized with available clinical data. Thus, we develop a parsimonious model that includes only general interactions between the tumor cells and the critical environmental factor in

**Table 1 Time to different stages of competitive release of T− cells for varying simulated treatment regimens**

| Time to progression | 5% T− | 90% T− | % Dose |
|---|---|---|---|
| *Representative patient #1—with "cheater" population* | | | |
| MTD | 3950 | 4621 | 100% |
| Metronomic | 4556 | 4950 | 24.1% |
| Adaptive | 8557 | Indefinite | 1.0% |
| *Representative patient #2—without "cheater" population* | | | |
| MTD | 323 | 784 | 100% |
| Metronomic | 323 | 784 | 24.1% |
| Adaptive | 323 | Indefinite | <1.0% |

No T− population is present before treatment in Patient #1 and only arises after therapy is given. Both MTD and the long induction of metronomic therapy results in a T− population appearing and it rapidly comprises the whole tumor. There is a slight lengthening of control in Patient #1 with metronomic therapy. Total dosing declines 24% relative to MTD. Under adaptive therapy, the substantial population of T+ cells delays the establishment of T− cells and provides durable control for a long period with minimal amounts of drug. Adaptive therapy prevents full competitive release of T−. Patient #2 has a population of T− in the tumor prior to first treatment. This is shown at time 323 in all cases. Both MTD and the long induction of metronomic therapy allows the T− population to quickly rise and experience full competitive release. Adaptive therapy in patient #2 prevents full competitive release of the T− population. The significantly lower percentage of drug given in the modeled adaptive therapies compared to the clinical trial percentages is due to the on/off pharmacokenetics in the model that leads to immediate reponses of cell populations. Patients likely experience more gradual changes and can only see a shift in therapy at most every 4 weeks

**Table 2 Summary of the diseases status of the 11 patients in the adaptive therapy trial**

| Subject ID | Gleason | Site of metastasis | Pre-abiraterone PSA | Abiraterone dose % |
|---|---|---|---|---|
| 1001 | 8(4 + 4) | Bone, soft tissue | 6.06 | 62% |
| 1002 | 8(4 + 4) | Bone | 58.57 | 55% |
| 1003 | 9 (5 + 4) | Bone, lymph node | 68 | 42% |
| 1005 | 7(4 + 3) | Lymph node | 95.86 | 87% |
| 1006 | 8(4 + 4) | Bone | 15.25 | 23% |
| 1007 | 6(3 + 3) | Bone | 109.4 | 21% |
| 1009 | 6(3 + 3) | Bone | 13.55 | 53% |
| 1010 | 7(3 + 4) | Bone | 17.33 | 45% |
| 1011 | 9(4 + 5) | Lymph node | 2.42 | 53% |
| 1012 | 8(4 + 4) | Bone | 4.17 | 21% |
| 1014 | 7(4 + 3) | Bone | 11.83 | 52% |
| Average cumulative abiraterone | | | | 47% |

The cumulative abiraterone dose as a percentage of SOC dose is shown for each patient in the trial

therapy—testosterone. We assume three competing phenotypes: (i) T+ cells requiring exogenous androgen; (ii) TP (testosterone producing) cells expressing CYP17A1 and producing testosterone; and (iii) T− cells that are androgen-independent and resistant to abiraterone.

**Mathematical model**. We use Lotka–Volterra (LV) competition equations to model the interactions among the T+, TP, and T− cell types, $i = 1, 2,$ and 3. The LV equations require parameterization of growth rates, $r_i$, carrying capacities, $K_i$, and the competition matrix,

$$\frac{dx_i}{dt} = r_i x_i \left( 1 - \frac{\sum_{(j=1)}^{3} a_{ij} x_j}{K_i} \right), \qquad (1)$$

where

| $a_{ij} =$ | | T+ | TP | T− |
|---|---|---|---|---|
| | T+ | $\alpha_{11}$ | $\alpha_{12}$ | $\alpha_{13}$ |
| | TP | $\alpha_{21}$ | $\alpha_{22}$ | $\alpha_{23}$ |
| | T- | $\alpha_{31}$ | $\alpha_{32}$ | $\alpha_{33}$ |

The intrinsic growth rates, $r_i$, were parameterized using measured doubling times of corresponding cell lines: ATCC@CRL-1740 LNCaP cell line, ATCC@CRL-2128 H295R cell line, and ATCC®CRL-1435 PC-3 cell lines for T+, TP, and T− cells, respectively. In this way, $r_i = [0.278, 0.355, 0.665]$. In vivo, these growth rates (units of per day) of cell lines are likely unrealistic and probably represent upper bounds. Hence, for subsequent simulations, we scaled each to 10% of these growth rates. Note that the intrinsic growth rates do not influence the equilibrium frequency of the three cell types in the absence of abiraterone. And, with abiraterone therapy, the qualitative cycling and effectiveness of adaptive therapy vs. standard of care is robust so long as the values of $r$ for the different cell types are of the same order of magnitude.

The carrying capacity of T+ in the absence of circulating testosterone (i.e., during ADT) derives entirely from "cheating"—i.e., utilizing the publically available testosterone produced by the TP cells. We assume that "cheating" is more profitable than the expense of producing testosterone and so we set $K_1 = 1.5 x_2$, allowing each TP cell to support the growth of 1.5 T+ cells.

We set the maximal carrying capacity of TP to $K_2 = 10,000$. We model the abiraterone therapy by decreasing this carrying capacity as it is unknown whether abiraterone causes cell death or quiescence. We let abiraterone reduce the carrying capacity of TP to 100. In this way, $K_2 \in [100, 10,000]$. Equally important, abiraterone inhibits the production of testosterone by the TP cells, and therefore it should diminish the ability of the TP cells to maintain the T+ "cheater" population. Thus, during administration of abiraterone, we set $K_1 = 0.5 x_2$.

As T− cells are unaffected by abiraterone and are not involved in the symbiotic "cheating", the carrying capacity remains constant at $K_3 = 10,000$. Model outcomes are not substantially affected by the actual magnitude of the $K$'s.

Much of the model's behavior hinges on the competition matrix, which acts to scale inter-cell type competitive effects. It is this matrix that characterizes the evolutionary game between the cancer cell types. Prior to and independent of the patient trial, we approximated the values of the competition matrix through a series of inequalities. These inequalities were derived from the literature[31] and professional judgment of prostate oncologists. Each competition coefficient represents the effect of an individual of type $j$ on the growth rate of type $i$. Since individuals of the same type are interchangeable we set the intra-type coefficients to one: $\alpha_{ii} = 1$. Furthermore, since there is both competition and some niche partitioning between the cell types with respect to association with the vasculature and the use of growth factors and growth factor pre-cursors, we assume that all coefficients are positive and < 1: $0 < \alpha_{ij} < 1$ for $i \neq j$. Two general rules determine the relative values of inter-cell type interactions; (1) T+ cells with no exogenous testosterone are in general the least competitive cell type, and (2) the competitive effect of T− cells is stronger on TP cells than on T+ cells. In this way $\alpha_{31} > \alpha_{21}$, $\alpha_{32} > \alpha_{12}$, $\alpha_{13} > \alpha_{23}$, $\alpha_{13} > \alpha_{12}$, $\alpha_{23} > \alpha_{21}$, and $\alpha_{32} > \alpha_{31}$.

There are 22 different rank orderings of the competition coefficients that satisfy these six inequality conditions. Li et al.[32], using a spatially-explicit, agent-based model provides a detailed analysis of the evolutionarily and ecologically stable communities that emerge from these 22 possible arrangements of the competition matrix. These are the communities that can evolve with ADT prior to abiraterone therapy. Furthermore, the different arrangements of the competition matrix may represent variability between different patients. For the following simulations and for any given rank ordering of the six inter-type competition coefficients, we used 0.4, 0.5, 0.6, 0.7, 0.8, and 0.9.

Best outcome responders: Twelve of the 22 communities promote an absence of T−, and high frequencies of both T+ and TP. In simulations, these tumor types respond well to therapy with large and relatively sustained drops in PSA. Under standard of care, T− will eventually emerge, increase to high frequency and cause progression. Under adaptive therapy (maintaining therapy when PSA is above 50% and ceasing therapy when PSA is below 50% of initial value) sustained cycles of PSA are possible. The total tumor burden declines with therapy and then increases when therapy is removed. We choose one representative patient (#1) from this category to explore model predictions.

Responders: Four matrix combinations result in low frequencies of T− at initiation of therapy. When simulating standard of care, the initial response is strong but unsustainable as T− quickly increases in frequency and results in progression. With adaptive therapy, progression still happens but TTP is greatly extended. We also choose another representative patient from this category (#2) to explore differences in treatment outcomes.

Non-responders: Six matrix combinations result in high equilibrium frequencies of T− ($\geq 20\%$). These competition matrices do not respond to therapy and do not result in even a 50% drop in tumor burden as measured by PSA. In simulations, with abiraterone the T− replace the TP and T+ cells quickly resulting in only a small initial response or no response at all. Empirically, some men do not show a response to abiraterone and are labeled as non-responders. In the trial that follows men who did not demonstrate an initial drop of at least 50% in their PSA were for ethical reasons excluded from receiving adaptive therapy with abiraterone. Thus, the actual trial will only involve men that possibly fall into the first two categories of matrix combinations.

**Table 3 Values of competition coefficients in representative patients responding to abiraterone therapy**

| | Representative patient #1 | | | | Representative patient #2 | | |
|------|------|------|------|------|------|------|------|
| | T+ | TP | T− | | T+ | TP | T− |
| T+ | 1 | 0.7 | 0.8 | T+ | 1 | 0.6 | 0.8 |
| TP | 0.4 | 1 | 0.5 | TP | 0.4 | 1 | 0.7 |
| T− | 0.6 | 0.9 | 1 | T− | 0.5 | 0.9 | 1 |

Variations in these parameters will alter the length of treatment cycles as shown in the simulations (Fig. 4) and confirmed by clinical data

To simulate different treatment strategies among men that should respond to abiraterone therapy, we chose two representative configurations (Table 3) of competition coefficients from the "responder" category above.

Because the direct correlation between tumor cell count and serum PSA is unknown, we simply assume that each cell produces one unit of PSA per unit time. We also assume that 50% of the PSA decays out of the serum each time step. In this way, the simulated serum PSA dynamics is given by:

$$\frac{dPSA}{dt} = \sum_{i=1}^{3} x_i - 0.5 \times PSA. \qquad (2)$$

For the two matrix combinations, we simulated four treatment regimens: first, the dynamics when no abiraterone treatment is given. Second, abiraterone was given continuously regardless of the simulated PSA value. Third, a predetermined intermittent on/off abiraterone treatment that matched the protocol in the published trial using intermittent ADT in castrate sensitive disease[20]. Finally, an adaptive therapy was created to match the clinical trial protocol by choosing a PSA value to begin abiraterone, removing abiraterone treatment once the PSA value drops to 50% of this value, and resuming abiraterone when PSA increased to the initial value.

For each simulation, we initialized a pre-abiraterone tumor. Within this tumor, the frequency of each cell type was set to what would be their equilibrium value for the LV model (the evolutionarily stable strategy, ESS) and the total population size of cancer cells and PSA level was set to 25% of what would be the untreated equilibrium. To simulate the various therapy regimes, we let the simulation run until the PSA hit 80% of the equilibrium. We let this be the value for progression to mCRPC and the trigger for the physician to initiate abiraterone therapy. When abiraterone is administered the carrying capacity of the TP cells drops to 100, and when abiraterone is withdrawn TP carrying capacity reverts to 10,000. We assume that patients progress radiographically when T− cells take over the tumor, and PSA levels remain constantly above 50% of the pre-abiraterone equilibrium level.

**Pilot clinical trial.** The theoretical analyses and model simulations provide a clear mechanism for the failure of prior intermittent trials (e.g., SWOG 9346) and identify a simple but evolutionarily-informed and patient-specific strategy for prolonging response to abiraterone. Based on the results of the simulations and building on prior translational studies[33, 34], we tested the model dynamics with IRB-approved trial in which abiraterone is administered to mCRPC patients through an adaptive therapy algorithm based on the evolutionary dynamics observed in the model.

Patient selection: Candidates for the study included patients with ECOG 0–2 performance status (PFS), adequate organ function and who had started abiraterone acetate plus prednisone as standard of care for progressive (PSA or imaging progression) mCRPC. Patients could be enrolled in the study after achieving a 50% or greater decline of their pre-abiraterone PSA levels. This patient population is similar to the AA-302[35] population except allowing ECOG 2 PFS, and prior treatment with Sipuleucel-T, and ketoconazole. Prior exposure to docetaxel was also allowed unless it was given during the castration resistant setting. Like the AA-302 trial[35], patients who took opioids for cancer-related pain were excluded.

Study design and treatment: This is a single institution investigator initiated pilot study funded by the Moffitt Cancer Center, Tampa, Florida. The protocol was approved by central IRB and monitored by Moffitt Cancer Center's protocol monitoring committee. Informed consent was obtained from all patients prior to enrollment in the trial. Each enrolled patient began on abiraterone (1000 mg by mouth daily; and prednisone) until achieving a > 50% decline in their baseline levels of PSA pre-abiraterone. Upon achieving this decline, abiraterone therapy was suspended.

Concomitantly and at the discretion of the investigator, patients would either stop prednisone or be tapered off prednisone. Patients were monitored every 4 weeks with a lab (CBC, COMP, LDH, and PSA) and clinic visit. Every 12 weeks, each patient received a bone scan, and a computed tomography (CT) of the abdomen and pelvis. Abiraterone plus prednisone were reinitiated when a patient's PSA increased to or above the pre-abiraterone baseline. Abiraterone therapy was stopped again after the patient's PSA declined to > 50% of his baseline PSA.

Each successive peak of PSA when abiraterone therapy was reinstated defined a complete cycle of adaptive therapy.

For patients who did not undergo surgical castration, GnRH analog treatment was continued to maintain castration levels of serum testosterone. Patients who did not achieve a 50% decline of their baseline PSA after restarting abiraterone remained on study until they developed radiographic progression while on abiraterone based on prostate cancer work group (PCWG)2 criteria[36]. Patients who developed radiographic progression while off abiraterone would restart abiraterone and remain on abiraterone until partial response was noted in the measurable lesions and stable disease was noted in the non-measurable lesions in the repeat bone scan, and abdominal and pelvic CT. These subjects were then allowed to stop abiraterone and reenter the adaptive therapy cycles. Patients are being followed until they develop radiographic progression or ECOG performance status deterioration while on abiraterone, whichever comes first. Of the 11 initial patients, 10 remain on trial and one has exhibited radiographic progression.

End points: The primary measurement end point was PSA response rate (defined as 50% decline of pre-abiraterone PSA) after completing two cycles of adaptive therapy. The secondary end points were median radiographic progression survival while on abiraterone and the median time to ECOG performance status deterioration. Radiographic progression-free survival was defined as freedom from death from any cause; freedom from progression in soft tissue lesions as measured with CT, defined as "progressive disease" according to modified Response Evaluation Criteria in Solid Tumors (RECIST) criteria; or progression on bone scan according to PCWG2 criteria.

**Data availability**. The details of the computational model and parameter estimates will be posted at https://github.com/cunninghamjj/Integrating-evolutionary-dynamics-into-treatment-of-mCRPC. The details of the pilot clinical trial are available at clinicaltrials.gov (NCT02415621).

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

## Acknowledgements

We thank Drs. Katerina Stankova, Yannick Viossat, and several anonymous reviewers for very helpful comments and critique. This work was supported by the James S. McDonnell Foundation grant, "Cancer therapy: Perturbing a complex adaptive system," NIH/National Cancer Institute (NCI) R01CA170595, Application of Evolutionary Principles to Maintain Cancer Control (PQ21), and NIH/NCI U54CA143970-05 (Physical Science Oncology Network (PSON)) "Cancer as a complex adaptive system". This work has also been supported in part by the Clinical Trials Core Facility at the H. Lee Moffitt Cancer Center and Research Institute, an NCI-designated Comprehensive Cancer Center (P30-CA076292), and the European Union's Horizon 2020 research and innovation program under the Marie Sklodowska-Curie grant agreement No 690817.

## Author contributions

R.A.G., J.S.B., J.Z., and J.J.C. contributed to the development of the evolutionary mathematical models, J.S.B. and J.J.C. performed the computational models, J.Z. conducted the clinical trial, J.S.B. performed the statistical analysis, R.A.G., J.S.B., J.Z., and J.J.C. wrote the manuscript.

## Additional information

**Competing interests:** The authors declare no competing financial interests.

