## [Peer Review File · Nature Communications]

Reviewers' comments:

Reviewer #1 (Remarks to the Author):

The authors describe an evolutionary game theory model that simulates the interaction between three cell types in metastatic castrate-resistant prostate cancer: cells which require testosterone, cells which produce testosterone themselves, and cells whose growth does not require testosterone. They model the competition and cooperation of these cell types with a Lotka-Volterra ODE system using an evolutionary game theory based interaction matrix and study four treatment strategies: maximum tolerated dose, metronomic therapy with a lengthy induction period, low-dose metronomic therapy, and adaptive therapy whereby treatment is stopped and reinitiated when the patient's PSA falls below 50% of the level at diagnosis. The authors then present findings from their clinical trial consisting of 11 patients that have been treated with abiraterone according to this adaptive therapy regimen. They are then monitored monthly until their PSA rises above their pre-abiraterone baseline, at which point they are restarted on abiraterone. Though the study has not concluded, the patients therein already have significantly increased median time of PSA or radiographic progression as compared to a pooled historic and contemporaneous control cohort.

There is significant tension between the two 'halves' of this paper: that is, the clinical and mathematical sides. The clinical part is well motivated, well cited and thoroughly described. The mathematical side however, is barely motivated, not placed in the context of the wider cancer and game theory literature in any way, and only partially described. The model has a staggering amount of possible complexity (even given its parsimony) and is only very scantily explored. We are told that the parameters are drawn from the 'literature and the judgement of prostate oncologists' - but are told no more. It is essentially used as an illustration of qualitative dynamics.

There is a somewhat concerning seeming self-contradiction as well. In both the introduction to and the discussion of the mathematical model, the authors state that adaptive therapy was modeled to match the clinical trial protocol by choosing a PSA value to begin abiraterone, and removing abiraterone treatment once the PSA value drops to 50% of this value; but then in the next breath they state "We examined model predictions in an IRB-approved trial in which abiraterone is administered to mCRPC patients through an adaptive therapy algorithm based on the above computer simulations". These two statements seem to be directly contradictory. Are the authors suggesting that the model informed trial design, or was the quantitative analysis done post-hoc?

The clinical data is quite impressive (and well analysed) and is a strong validation for the incorporation of evolutionary dynamics into clinical decision-making. However, the cohort is quite small and the endpoints of the study have not yet been yet. In the abstract, the authors state "Interim analysis of a pilot clinical trial testing model predictions showed that one patient progressed but the remaining 10 of 11 patients have stably oscillating tumor burdens [lines 43 - 45]." In the results of the clinical trial, the authors state that "One patient developed PSA and radiographic progression at month 11. Two patients have exhibited PSA progression at 21 and 28 months but remain on trial with no radiographic progression. [lines 283 - 285]." This radiographic progression is not shown in Figure 5.

In conclusion, I think that there are two very exciting pieces of research happening here, but neither is mature on its own. The clinical trial results seem like they will (if they continue as they have) change practice, and outcomes for many men. The idea that evolutionary dynamics should be incorporated into clinical trial design also is very appealing. The model however, is unsatisfactorily presented - the authors state that there are 22 meaningful dynamical regimes, but only a small portion are considered quantitatively. The parameters are invented (which is fine in some circumstances, but this mandates even more sharply a proper parameter sensitivity analysis which is entirely lacking). The model is also not placed in any way into the larger context of the evolutionary game theory and cancer.

I have a number of specific major comments in addition to the more vague concerns aired above.

- what is reference 27? it is referenced, but does not exist in the bibliography (and is there a 25 and 26?)
- figure 1 would benefit from also have an example temporal plot of the data shown in the cartoon to ease the reader into later plots
- there are growth rates cited for the r_i parameter - who measured these? the authors? citation? (there is also a typo in the statement of the r vector 0.0.xx)
- i've already stated this, but it bears repeating: there is a great intro into the clinical trial background, but none into EGT and cancer. This is not a new field. Indeed, there are papers describing hormone therapy timing and duration in prostate cancer which should be mentioned at the very least
- I find the parameter choices for carrying capacity (and the interaction matrix) quite arbitrary and there is no attempt at a sensitivity analysis

Reviewer #2 (Remarks to the Author):

The authors have translated their pre-clinical results, controlling therapeutic resistance, to the clinic for the first time. The results are striking. Even though this is a pilot study, and so the study size is small, the results are highly significant, and so unlikely to be due to chance. Furthermore, the community should see these results so that there is reason and justification for investing in the larger clinical trials that would validate the results. To understand the significance of these results, we have to understand that oncology has essentially had no plan for how to deal with therapeutic resistance. We either wait for one drug to fail, and then try a second or third line therapy, typically with diminishing efficacy, until we run out of drugs. Or we combine drugs in the hope that, like combination therapy in HIV, we can make resistance so unlikely as to never occur. Of course, it does occur, typically only a few months later than it would have occurred with single drug therapy (though with some exceptions, primarily in childhood cancers and blood cancers that have insufficient heterogeneity to harbor resistant clones). Gatenby et al. have borrowed ideas from pest management in agriculture to control resistance in cancer. It appears to be working. This could be the most important advance in oncology in decades.

Note also that this is a successful example of personalized medicine. The models have to be fit to the individual's data, in order to infer what is happening in that tumor and respond effectively to the dynamics of that specific tumor. In comparison, both intermittent and metronomic therapy, administered without regard to how the tumor is responding, fails in simulation. This reflects what was found in pest management – it is information intensive – you have to keep monitoring the pests (tumor) to respond appropriately.

The analysis of SWOG trial 9346 with intermittent therapy is illuminating, and would be worthy of publication by itself (with more details).

Critiques:

Lines 139-140: deriving the growth rates from cell lines in vitro makes me nervous. Are the results sensitive to these parameters?

Lines 144-154: I'm uncomfortable with the ad hoc choices of K 's and the effects of TP on T+ cells. Are the results sensitive to these choices? I'd like to see a sensitivity analysis of the model.

Line 164: Why is the competitive effect of T- cells stronger on TP than on T+ cells? Is it because

only T- and TP cells can occupy niches that lack testosterone?

Line 188: It seems that the adaptive therapy protocol essentially matches the AT-2 algorithm from the authors STM paper that didn't work. Why did they go with AT-2 and not AT-1 that did work in the pre-clinical models?

Are the modeling tools and dosing recommendations available to researchers who would like to replicate this study?

Minor comments:

Line 71: shouldn't "proliferative advantage" be "competitive advantage" since the advantage might be due to some characteristic other than proliferation (e.g. survival)?

Line 76: "remained elusive" I think is a bit misleading, as it implies that people have tried but there have been problems. If I understand correctly, the current manuscript represents the first time anyone has tried, right?

Lines 33 and 84: Why does the abstract say most patients progress within ~16 months while line 84 says median time to progression is 5.8-11 months? Since the abstract later says median TTP is >24 months in this study, shouldn't line 33 refer to the 5.8-11 month statistic? I see that the complicating issue is PSA vs. scan progression. I think it is OK to only report one of those in the abstract (e.g., scan progression) but it should be made explicit what you are talking about.

Line 117: "system" should be "systems"

Line 125: This should be rewritten to make it obvious what TP stands for "Testosterone producing (TP) cells expressing CYP17A1..."

Line 143: "then" should be "than"

It should be mentioned somewhere that adaptive therapy only appears to work if there is an initial response to the drug. This explains the entry criteria of >50% shrinkage on abiraterone.

Line 213: What does prednisone do? Why is it being given with abiraterone?

Table 1: Is there a word missing in the upper left corner "Time to progression to", to what?

Table 1: I gather that the reason there aren't entries for the last row of the table for "Adaptive with cheaters" was that CR never reached. That isn't clear from the table or the table legend. I'd suggest filling it in with >####, where #### is the number when the simulation was terminated.

Figure 5: The one patient in the upper panel with scan progression is not visually clear. The x symbol should either be changed or placed on top of the triangle. I'd also like to see this in a Kaplan-Meier survival curve format with 95% confidence intervals.

Lines 44 and 299 give different stats: mean vs. median TTP can be no less than 24 and 27 months, respectively. I don't know the clinical trial literature well enough to know what is preferred, but I'm guessing median.

Lines 311-312 is a little confusing. It sounds like the chance that adaptive therapy has mean TTP < 17 months is 98%.

Lines 324-334: It seems like a Cox regression would take into account time to progression and be

a more powerful statistical test, if the prerequisites of the regression are met. But, given that the more conservative (weaker) tests are highly significant, I don't think this is required.

Line 356: The authors should detail the proposed improvements in therapy, so that anyone wanting to follow-up on these results has the benefit of those results.

Response to Referees' comments:

Reviewer #1 (Remarks to the Author):

The authors describe an evolutionary game theory model that simulates the interaction between three cell types in metastatic castrate-resistant prostate cancer: cells which require testosterone, cells which produce testosterone themselves, and cells whose growth does not require testosterone. They model the competition and cooperation of these cell types with a Lotka-Volterra ODE system using an evolutionary game theory based interaction matrix and study four treatment strategies: maximum tolerated dose, metronomic therapy with a lengthy induction period, low-dose metronomic therapy, and adaptive therapy whereby treatment is stopped and reinitiated when the patient's PSA falls below 50% of the level at diagnosis. The authors then present findings from their clinical trial consisting of 11 patients that have been treated with abiraterone according to this adaptive therapy regimen. They are then monitored monthly until their PSA rises above their pre-abiraterone baseline, at which point they are restarted on abiraterone. Though the study has not concluded, the patients therein already have significantly increased median time of PSA or radiographic progression as compared to a pooled historic and contemporaneous control cohort.

There is significant tension between the two 'halves' of this paper: that is, the clinical and mathematical sides. The clinical part is well motivated, well cited and thoroughly described. The mathematical side however, is barely motivated, not placed in the context of the wider cancer and game theory literature in any way, and only partially described. The model has a staggering amount of possible complexity (even given its parsimony) and is only very scantily explored. We are told that the parameters are drawn from the 'literature and the judgement of prostate oncologists' - but are told no more. It is essentially used as an illustration of qualitative dynamics.

We very much appreciate and agree with the reviewer's comments. Integrating mathematical models into clinical oncology is both difficult to achieve and to integrate within a manuscript. Since we have found that most reviewers of such papers are clinically focused, we have tended to limit the modeling details that are presented. This is the case here as another mathematical manuscript (You, et al – now referenced in this paper) extensively explores the underlying model dynamics has been recently accepted for publication in the Journal of Theoretical Biology. That said, we have edited the current manuscript to describe the modeling assumptions, parameter estimates, and simulation outcomes. We have expanded and clarified the methods of the modelling, and we have added the details necessary for anyone to explore and reproduce the results. We have included 8 references to align with prior work on cancer evolutionary game theory, to support parameters used in the model and to give background to the modeling strategy itself.

There is a somewhat concerning seeming self-contradiction as well. In both the introduction to and the discussion of the mathematical model, the authors state that adaptive therapy was modeled to match the clinical trial protocol by choosing a PSA value to begin abiraterone, and removing abiraterone treatment once the PSA value drops to 50% of this value; but then in the next breath they state "We examined model predictions in an IRB-approved trial in which abiraterone is

administered to mCRPC patients through an adaptive therapy algorithm based on the above computer simulations". These two statements seem to be directly contradictory. Are the authors suggesting that the model informed trial design, or was the quantitative analysis done post-hoc?

Excellent point. We have amended the manuscript to be clearer about the lock-step approach that was used to develop the model and the clinical trial. As noted above, integrating theoretical models with clinical trial design requires a prolonged period of multidisciplinary discussion. The mathematical model was developed before the trial was designed based on the initial discussions that focused on the likely mechanism of action for abiraterone and strategies that were used by cells to become resistant. The three population model and the pre-treatment trade-of matrix were designed during this part of the work. Once the models were developed, we gained confidence in them when they successfully predicted outcomes for standard-of-care continuous abiraterone dosing as well as the prior unsuccessful (SWOG) trials using intermittent therapy. The specific adaptive treatment strategies then tested by the model were limited to approaches (i.e. the on/off pattern) that were felt to be clinically feasible based on additional discussion. Model predictions demonstrated that a range of adaptive therapy strategies would prolong the time to progression and permit lower drug doses. The oncologist (Dr. Zhang) decided on the specific criteria for starting or stopping abiraterone based on PSA measurements. Thus, the models provided the general strategy but did not dictate the specific details of the protocol or clinical decisions during treatment.

The clinical data is quite impressive (and well analysed) and is a strong validation for the incorporation of evolutionary dynamics into clinical decision-making. However, the cohort is quite small and the endpoints of the study have not yet been yet. In the abstract, the authors state "Interim analysis of a pilot clinical trial testing model predictions showed that one patient progressed but the remaining 10 of 11 patients have stably oscillating tumor burdens [lines 43 - 45]." In the results of the clinical trial, the authors state that "One patient developed PSA and radiographic progression at month 11. Two patients have exhibited PSA progression at 21 and 28 months but remain on trial with no radiographic progression. [lines 283 - 285]." This radiographic progression is not shown in Figure 5. ***It should be, but perhaps hard to see?

The reviewer is correct and we thank him/her for pointing this out. We forgot to mark scan progression on subject 1010 and have amended the figure in our revised manuscript.

In conclusion, I think that there are two very exciting pieces of research happening here, but neither is mature on its own. The clinical trial results seem like they will (if they continue as they have) change practice, and outcomes for many men. The idea that evolutionary dynamics should be incorporated into clinical trial design also is very appealing. The model however, is unsatisfactorily presented - the authors state that there are 22 meaningful dynamical regimes, but only a small portion are considered quantitatively. The parameters are invented (which is fine in some circumstances, but this mandates even more sharply a proper parameter sensitivity analysis which is entirely lacking). The model is also not placed in any way into the larger context of the evolutionary game theory and cancer.

We accept the criticism and have amended the manuscript to address these concerns. As noted above, we have added 8 references all related to the mathematical model. We have also

significantly expanded the methods and results sections that deal with the mathematical model to provide more details regarding the model and simulations. The modelling and clinical work has been highly multidisciplinary. Additional discussion of the more technical aspects of the model can be found in another manuscript recently accepted for the Journal of Theoretical Biology. We have appended this manuscript should the reviewer be interested. In this manuscript, the three cell types are analyzed as a spatially-explicit agent based model based on the 22 configurations of the competition matrix (in the JTB paper presented as a modified but analogous payoff matrix). The JTB manuscript does not include therapy but analyzes in detail the 22 configurations and various critical assumptions.

I have a number of specific major comments in addition to the more vague concerns aired above.

- what is reference 27? it is referenced, but does not exist in the bibliography (and is there a 25 and 26?)

We apologize and have fixed the reference numbers in addition to adding new references.

- figure 1 would benefit from also have an example temporal plot of the data shown in the cartoon to ease the reader into later plots

This is a great idea and we have amended Figure 1 accordingly

- there are growth rates cited for the r_i parameter - who measured these? the authors? citation? (there is also a typo in the statement of the r vector 0.0.xx)

We have amended the manuscript to make this clearer. The measured growth rates are from culture conditions and are viewed as the upper bound of in vivo growth rates. This is now discussed in more details including the range of parameters that were simulated. In particular we re-ran the simulations with maximum growth rates (r) set to 10% of the cell culture rates.

- i've already stated this, but it bears repeating: there is a great intro into the clinical trial background, but none into EGT and cancer. This is not a new field. Indeed, there are papers describing hormone therapy timing and duration in prostate cancer which should be mentioned at the very least

- I find the parameter choices for carrying capacity (and the interaction matrix) quite arbitrary and there is no attempt at a sensitivity analysis

We agree and welcome the chance to be more expansive. Thus, we have extensively revised the manuscript to provide a broader coverage of our simulation results to demonstrate which are robust which are not.

Reviewer #2 (Remarks to the Author):

The authors have translated their pre-clinical results, controlling therapeutic resistance, to the clinic for the first time. The results are striking. Even though this is a pilot study, and so the study size is small, the results are highly significant, and so unlikely to be due to chance. Furthermore, the community should see these results so that there is reason and justification for investing in the larger clinical trials that would validate the results. To understand the significance of these results, we have to understand that oncology has essentially had no plan for how to deal with therapeutic resistance. We either wait for one drug to fail, and then try a second or third line therapy, typically with diminishing efficacy, until we run out of drugs. Or we combine drugs in the hope that, like combination therapy in HIV, we can make resistance so unlikely as to never occur. Of course, it does occur, typically only a few months later than it would have occurred with single drug therapy (though with some exceptions, primarily in childhood cancers and blood cancers that have insufficient heterogeneity to harbor resistant clones). Gatenby et al. have borrowed ideas from pest management in agriculture to control resistance in cancer. It appears to be working. This could be the most important advance in oncology in decades.

Note also that this is a successful example of personalized medicine. The models have to be fit to the individual's data, in order to infer what is happening in that tumor and respond effectively to the dynamics of that specific tumor. In comparison, both intermittent and metronomic therapy, administered without regard to how the tumor is responding, fails in simulation. This reflects what was found in pest management – it is information intensive – you have to keep monitoring the pests (tumor) to respond appropriately.

This is a good point and we have edited the manuscript to reflect this.

The analysis of SWOG trial 9346 with intermittent therapy is illuminating, and would be worthy of publication by itself (with more details).

Critiques:

Lines 139-140: deriving the growth rates from cell lines in vitro makes me nervous. Are the results sensitive to these parameters?

Lines 144-154: I'm uncomfortable with the ad hoc choices of K 's and the effects of TP on T+ cells. Are the results sensitive to these choices? I'd like to see a sensitivity analysis of the model.

Both of the above points are excellent and similar to comments by reviewer 1. As noted in our response above, we have now added considerable text to the methods and results sections to clarify our choice of parameters, the range of parameter values that were tested along with a sensitivity analysis. Also as noted above, we have separately submitted a far more detailed analysis of the modeling results for the mathematically inclined readers. The manuscript has been accepted for publication in Journal of Theoretical Biology. A preprint is appended to this document.

Line 164: Why is the competitive effect of T- cells stronger on TP than on T+ cells? Is it because only T- and TP cells can occupy niches that lack testosterone?

This is certainly a reasonable hypothesis. In the current manuscript we have presented far more details on the range of parameter estimates and our reasoning behind them. Much of our modeling approach used simple inverse problem reasoning. Since virtually all prostate cancers respond to androgen deprivation therapy, we assume that the T+ population is initially dominant and thus the overall fittest phenotype in an environment with exogenous testosterone. Since about 62% of castrate-resistant prostate cancers respond to abiraterone, we assume that in most (but not all) mCRPC tumors, the TP population is fitter than the T- phenotype. We agree that the reviewer's proposed mechanism is a good explanation for the observed variations in fitness but this will require additional studies to confirm. We have amended the manuscript to be clearer on these methods.

Line 188: It seems that the adaptive therapy protocol essentially matches the AT-2 algorithm from the authors STM paper that didn't work. Why did they go with AT-2 and not AT-1 that did work in the pre-clinical models?

Yes, in the pre-clinical models, dose skipping (AT2) did not work very well compared to continuous dose modulation. The reason was that in the period between doses, tumor growth could be so rapid that we could not regain control with subsequent treatment. We worried about this in the design of the clinical trial. This was one of the reasons we spent much time on the math models. It appears that there are several important dynamics in play here. First, human tumors grow far slower than the mouse tumors we investigated (which could double in as few as 3 days). Thus, we did not see rapid tumor growth when we withdrew abiraterone – in fact it was generally slower than we expected. In addition, the “cheating” dynamics between the TP and T+ cells added to the treatment-sensitive populations and likely increased the suppression of the T- cells. Nevertheless, this continues to be a concern in designing new clinical trials and is always a point that is addressed in the protocol design.

Are the modeling tools and dosing recommendations available to researchers who would like to replicate this study?

A great point. We have significantly increased the material presented in methods and results regarding the mathematical model and computer simulations. We believe that readers can readily reproduce this methodology based on the material presented.

Minor comments:

Line 71: shouldn't “proliferative advantage” be “competitive advantage” since the advantage might be due to some characteristic other than proliferation (e.g. survival)?

Agreed – we have changed this.

Line 76: “remained elusive” I think is a bit misleading, as it implies that people have tried but there have been problems. If I understand correctly, the current manuscript represents the first time anyone has tried, right?

Thank you. We viewed prior efforts to use intermittent treatment as attempts to exploit evolutionary dynamics and wanted to acknowledge those trials. The line was changed to “efforts

to translate evolutionary dynamics into a clinical setting have generally used informal, non-quantitative approaches to define the underlying Darwinian dynamics.”

Lines 33 and 84: Why does the abstract say most patients progress within ~16 months while line 84 says median time to progression is 5.8-11 months? Since the abstract later says median TTP is >24 months in this study, shouldn't line 33 refer to the 5.8-11 month statistic? I see that the complicating issue is PSA vs. scan progression. I think it is OK to only report one of those in the abstract (e.g., scan progression) but it should be made explicit what you are talking about.

The problem here is that radiographic progression usually occurs later than PSA progression. So, the time quoted in the abstract is radiographic progression and the time noted in the text refers to PSA progression. Furthermore, both times to progression (i.e PSA and radiographic) tended to be significantly shorter when there has been prior treatment with Docetaxel. Since patients with prior treatment with Docetaxel were eligible for our trial. This is a relevant number. This is all admittedly quite confusing but is consistent with the way data in prostate cancer clinical trials are presented. In the statistical analysis section of the paper, we are quite rigorous and conservative about using data only from our own patients and those in published studies who had not received prior Docetaxel. We have tried to clarify these points in the abstract and the manuscript.

Line 117: "system" should be "systems"

Fixed

Line 125: This should be rewritten to make it obvious what TP stands for "Testosterone producing (TP) cells expressing CYP17A1"

Line 143: "ethen" should be "ethan"

This section has been extensively rewritten and I cannot now find the typo

It should be mentioned somewhere that adaptive therapy only appears to work if there is an initial response to the drug. This explains the entry criteria of >50% shrinkage on abiraterone.

Done

Line 213: What does prednisone do? Why is it being given with abiraterone?

Administration of prednisone during hormonal therapy is standard practice in treatment of prostate cancer. It was part of the treatment regimen in all of the published studies used in our manuscript. Secreted levels of ACTH increase in response to decreased levels of cortisol due to CYP17 complex inhibition by abiraterone. Coadministration of prednisone suppresses the ACTH drive and reduces the incidence and severity of mineralocorticoid excess adverse reactions like hypertension, hypokalemia, and fluid retention. All of the patients in the contemporary cohort and in the published trials used for comparison received identical doses prednisone.

Table 1: Is there a word missing in the upper left corner "Time to progression to", to what?

Good observation. We have corrected this.

Table 1: I gather that the reason there aren't entries for the last row of the table for "Adaptive with cheaters" was that CR never reached. That isn't clear from the table or the table legend. I'd suggest filling it in with >####, where #### is the number when the simulation was terminated.

Good point. We have added the number of cycles prior to simulation termination to the caption.

Figure 5: The one patient in the upper panel with scan progression is not visually clear. The x symbol should either be changed or placed on top of the triangle. I'd also like to see this in a Kaplan-Meier survival curve format with 95% confidence intervals.

Kaplan-Meier survival curves are not terribly informative at this stage. We could do so for the contemporaneous controls where most patients have progressed. But with just one radiographic progression among our 11 patients and varied lengths of time on trial for the rest, it is not possible to make a rigorous or convincing plot, other than to have a single step early and a straight line afterwards. This will become more relevant as more men progress or remain progression free beyond 25 – 30 months.

Lines 44 and 299 give different stats: mean vs. median TTP can be no less than 24 and 27 months, respectively. I don't know the clinical trial literature well enough to know what is preferred, but I'm guessing median.

This is a good point. By convention, clinical trials uniformly report the median results in a study cohort. As noted in our statistics section, the mean is not available in many trials and has to be estimated. The mean value becomes necessary for making statistical comparisons with our small sample size, as we have been able to do. Other than sign tests or chi-square tests of heterogeneity there are no rigorous statistical tests for comparing medians in the absence of other statistical moments. Our statistical section now includes both mean and median values. And in inferring the mean of the large published trial from the median we have used a very conservative transformation to insure we do not underestimate the value when comparing to the mean of our trial with its small sample size.

Lines 311-312 is a little confusing. It sounds like the chance that adaptive therapy has mean TTP < 17 months is 98%.

We agree this is a confusing sentence and is meant to convey that this is highly improbable. Since the preceding sentence states the same thing more clearly we have simply eliminated the confusing statement.

Lines 324-334: It seems like a Cox regression would take into account time to progression and be a more powerful statistical test, if the prerequisites of the regression are met. But, given that the more conservative (weaker) tests are highly significant, I don't think this is required.

We appreciate this point. In performing a Cox regression or proportional hazard analysis we would have to make estimates for time to progression or lack thereof of the 10 men that have not yet progressed. If we assume that all progress at the time of writing the model gives essentially an identical statistical result to what we have presented. If agreeable, we prefer the more conservative tests to insure confidence in our conclusions of superiority from a small sample size. But, we can perform the Cox model if desired.

Line 356: The authors should detail the proposed improvements in therapy, so that anyone wanting to follow-up on these results has the benefit of those results.

We hope that sufficient detail is now given for both the therapy and the model to permit others to expand and replicate the theory and practice. Furthermore, the models can and should be improved through additional data currently being investigated such as circulating tumor cells, CTCs, and image analytic methods. The sentence has been amended to make that clearer.

Spatial vs. non-spatial eco-evolutionary dynamics in a tumor growth model

Li You^a, Joel S. Brown^{c,d}, Frank Thuijsman^a, Jessica J. Cunningham^d,
Robert A. Gatenby^{d,e}, Jingsong Zhang^f, Kateřina Staňková^{a,b,g}

^aDepartment of Data Science and Knowledge Engineering, Maastricht University, Maastricht, The Netherlands

^bDelft Institute of Applied Mathematics, Technical University Delft, Delft, The Netherlands

^cDepartment of Biological Sciences, University of Illinois at Chicago, Chicago, Illinois, USA

^dDepartment of Integrated Mathematical Oncology, Moffitt Cancer Center & Research Institute, Tampa, Florida, USA

^eDepartment of Diagnostic Imaging and Interventional Radiology, Moffitt Cancer Center & Research Institute, Tampa, Florida, USA

^fDepartment of Genitourinary Oncology, Moffitt Cancer Center & Research Institute, Tampa, Florida USA

Abstract

Metastatic prostate cancer is initially treated with androgen deprivation therapy (ADT). However, resistance typically develops in about 1 year – a clinical condition termed metastatic castrate-resistant prostate cancer (mCRPC). We develop and investigate a spatial game (agent based continuous space) of mCRPC that considers three distinct cancer cell types: 1) those dependent on exogenous testosterone (T^+), 2) those with increased CYP17A expression that produce testosterone and provide it to the environment as a public good (T^P), and 3) those independent of testosterone (T^-). The interactions within and between cancer cell types can be represented by a 3×3 matrix. Based on the known biology of this cancer there are 22 potential matrices that give roughly three major outcomes depending upon the absence (good prognosis), near absence or high frequency (poor prognosis) of T^- cells at the evolutionarily stable strategy (ESS). When just two cell types coexist the spatial game faithfully reproduces the ESS of the corresponding matrix game. With three cell types divergences occur, in some cases just two strategies coexist in the spatial game even as a non-spatial matrix game supports all three. Discrepancies between the spatial game and non-spatial ESS happen because different cell types become more or less clumped in the spatial game – leading to non-random assortative interactions between cell types. Three key spatial scales influence the distribution and abundance of cell types in the spatial game: i. Increasing the radius at which cells interact with each other can lead to higher clumping of each type, ii. Increasing the radius at which cells experience limits to population growth can cause densely packed tumor clusters in space, iii. Increasing the dispersal radius of daughter cells promotes increased mixing of cell types. To our knowledge the effects of these spatial scales on eco-evolutionary dynamics have not been explored in cancer models. The fact that cancer interactions are spatially explicit and that our spatial game of mCRPC provides in general different outcomes than the non-spatial game might suggest that non-spatial models are insufficient for capturing key elements of tumorigenesis.

Keywords: Prostate Cancer, Evolutionary Game Theory, Spatial Game, Non-spatial Game

1. Introduction

In cancer biology, tumors are viewed as complex ecosystems consisting of cancer cells, normal cells, blood vasculature, inter-cellular spaces, and various nutrients such as oxygen and glucose [1,

* Corresponding author

Email address: k.stankova@maastrichtuniversity.nl (Kateřina Staňková)

Reviewers' comments:

Reviewer #1 (Remarks to the Author):

General comments:

I very much appreciate the authors' careful attention to my comments and those of my co-reviewer. This paper is vastly improved. I do still have some reservations and suggestions for improvement that I would like to see amended before publication. Most important is that there is still tension between what came first: the model or the trial. This question is answered carefully and well in the authors' response to the reviewers, but there remains several places in the manuscript where this is unclear (in particular the sentences beginning on line 230 and 246 seem to be in disagreement with one another). Please state this clearly in the paper as you have done in the response letter. This will accomplish the clarification that I want, and also highlight to the reader how multi-disciplinary projects like this progress.

As it stands, the model simply has only qualitatively affected the trial. You state: ``Thus, the models provided the general strategy but did not dictate the specific details of the protocol or clinical decisions during treatment.' This is not clear in the paper - also, it makes one wonder why we need any of the quantitative model results at all. I would submit that we DO need the quantitative results, and would like this to shine through for the reader as well.

Maybe a short discussion of how the model could guide actual therapy rather than simply as a cartoon? Could patient response be used to individually parameterize models to guide patients on a personal level, like the work of Rockne and Swanson in glioblastoma (PMID: 20484781 out of many examples) or Werner et al. (PMID: 26833122) or any of many other examples? Could individual tumor biopsy results be used in place of the cell lines - like in the work of Drs. Silva and Gatenby in Myeloma (PMID: 28400475)? In short, beyond providing a cartoon of evolution, what more can we learn from this model? I think a few sentences could assuage my concerns here quite easily, and would be a nice place to cite the many perspective pieces that are cited earlier (like in the second paragraph where perspective pieces are cited instead of primary work).

I very much appreciate the attention paid to rigorous and reproducible presentation of the model. There are only a few small things that have been left out that remain. Line 193 makes a claim about variance and coexistence changes with parameter changes which is unsupported. I do not think it needs to be supported in the main text, but if this claim is stated, the evidence for it should be presented in supplemental material. The same comment is true about changing K not changing qualitative results (lines 158-169). My co-reviewer made several comments about K, so I don't think this is unwarranted.

Specific issues:

* I appreciate the effort to improve Figure 1, however, it is currently impossible to read the legend in the new sub-figures (temporal dynamics). There seem to be five species depicted despite there only being two on the left side of the figure. Using two species and keeping the same orange/blue colors as the left side of the figure would greatly help readability. The caption could then be updated to better describe both sides of the figure simultaneously. For clinical readers who haven't seen dynamical plots like this, this figure is necessary as a segue to the ones that follow of the EGT model.

* That the model can account for both slow- and fast-cycling PSA dynamics is admirable, but there is significant mismatch between the simulations in Figure 4 and the patient data. This, I believe, is partly due to PSA and tumor cell populations dropping instantaneously in the model, but dropping rather slower in patients. This could be remedied by including indicators of when patient data was measured in the plot. Stars or x's or something.

* Table 1 could be better presented if "Adaptive without cheaters" and "Adaptive with cheaters" were replaced with "Adaptive" and the status of cheater cells was placed in the "Representative Patient..." area. Additionally, I would suggest replacing the "---" with "indefinite" or some other word because as the table is presented now, at first it seems that those therapies were not attempted rather than the excellent responses they achieved.

* Table 1 also shows the large difference between the model's predicted percent dose and the clinical data's percent dose. The representative patients in Table 1 received less than 2% of the MTD, whereas the actual patients have so far received 47%. I suspect this is due to instantaneous drops in tumor cell populations in the model.

* Lines 324 - 327 describe Table 1 and state that "adaptive therapies provide significant increase in time to progression under any initial tumor condition." Only two representative patients are shown in Table 1 which are, presumably the same representative patients from line 220. This seems too few initial conditions to present to make such a sweeping claim, especially when the two representative patients' sets competition coefficients are in the same category of responder.

Reviewer #2 (Remarks to the Author):

All of my critiques have been adequately addressed with two small exceptions:

The authors say they changed line 76 to: "efforts to translate evolutionary dynamics into a clinical setting have generally used informal, non-quantitative approaches to define the underlying Darwinian dynamics." but that change does not appear to have been made.

I found a typo: Line 124 should read "Prostate cancer, like all tumors, is an open..."

Response to Reviewers' comments:

We thank the reviewers for their meticulous reading of the manuscript and their consistently helpful suggestions. We have now revised the manuscript in response to their comments. Primarily we have added material to clarify the role of the model in both the justification and design of our clinical trial. We have also made a number of revisions in the modeling details, added 3 supplemental tables, and revised the figures for greater clarity.

Below is our specific response to the reviewers' comments.

Response to reviewers

Reviewer 1

Comment 1: Most important is that there is still tension between what came first: the model or the trial. This question is answered carefully and well in the authors' response to the reviewers, but there remains several places in the manuscript where this is unclear (in particular the sentences beginning on line 230 and 246 seem to be in disagreement with one another). Please state this clearly in the paper as you have done in the response letter. This will accomplish the clarification that I want, and also highlight to the reader how multi-disciplinary projects like this progress.

Response 1: We thank the reviewer for his meticulous and thoughtful comments. The sentence in line 230 refers to modeling efforts that match the protocol used in a prior, published trial using intermittent Androgen Deprivation Therapy for castrate-sensitive prostate cancer. It does not refer to the trial presented in this manuscript. We have rewritten the sentence to clarify this. The sentence in 246 is correct and has not been changed. We have also added the following to the abstract to make the sequence of events clear: "The successful strategy identified in the model simulations was tested in a

pilot clinical trial. Interim analysis of the first 11 patients accrued to the trial found 10 still maintaining stably oscillating tumor burdens.”

Comment 2: As it stands, the model simply has only qualitatively affected the trial. You state: “Thus, the models provided the general strategy but did not dictate the specific details of the protocol or clinical decisions during treatment.” This is not clear in the paper - also, it makes one wonder why we need any of the quantitative model results at all. I would submit that we DO need the quantitative results, and would like this to shine through for the reader as well. Maybe a short discussion of how the model could guide actual therapy rather than simply as a cartoon? Could patient response be used to individually parameterize models to guide patients on a personal level, like the work of Rockne and Swanson in glioblastoma (PMID: 20484781 out of many examples) or Werner et al. (PMID: 26833122) or any of many other examples? Could individual tumor biopsy results be used in place of the cell lines - like in the work of Drs. Silva and Gatenby in Myeloma (PMID: 28400475)? In short, beyond providing a cartoon of evolution, what more can we learn from this model? I think a few sentences could assuage my concerns here quite easily, and would be a nice place to cite the many perspective pieces that are cited earlier (like in the second paragraph where perspective pieces are cited instead of primary work).

Response 2: We clearly have failed to convey this point accurately. In planning this clinical trial we were repeatedly criticized because prior attempts at intermittent therapy in prostate cancer had failed. Thus, most clinical investigators viewed this as question that had been settled. Thus, the models were, in fact, essential both to understand the reasons for the prior failure and to clearly identify an alternative that would work as well as providing a convincing mechanism for the reason that it would succeed while others failed. Thus, the models played a critical role not only in the design of the trial but in the willingness of the SRC and IRB to approve it. To clarify this, in addition to the changes above, we have rewritten the introduction to the clinical trial as follows:

“The theoretical analysis and model simulations provided a clear mechanism for the failure of prior intermittent trials (SWOG 9346) and identified a simple but evolutionarily-informed and patient-specific strategy to prolong response to abiraterone. Based on the results of the of the simulations and building on prior translational studies^{28, 29}, we tested the model dynamics into an IRB-approved trial in which abiraterone is administered to mCRPC patients through an adaptive therapy algorithm based on the evolutionary dynamics observed in silico.”

Note that we have added two of the suggested references.

Also, we point out the following sentence in the Discussion:

"Efforts to translate evolutionary dynamics into a clinical setting have generally used informal, non-quantitative approaches to define the underlying Darwinian dynamics."

Comment 3: I very much appreciate the attention paid to rigorous and reproducible presentation of the model. There are only a few small things that have been left out that remain. Line 193 makes a claim about variance and coexistence changes with parameter changes which is unsupported. I do not think it needs to be supported in the main text, but if this claim is stated, the evidence for it should be presented in supplemental material. The same comment is true about changing K not changing qualitative results (lines 158-169). My co-reviewer made several comments about K, so I don't think this is unwarranted.

Response 3: Thank you. We have extensively rewritten the modeling methods section. We have also added Supplemental Tables 1, 2, and 3. This includes

clarifying the values used for the carrying capacity K .

Specific issue 1: I appreciate the effort to improve Figure 1, however, it is currently impossible to read the legend in the new sub-figures (temporal dynamics). There seem to be five species depicted despite there only being two on the left side of the figure. Using two species and keeping the same orange/blue colors as the left side of the figure would greatly help readability. The caption could then be updated to better describe both sides of the figure simultaneously. For clinical readers who haven't seen dynamical plots like this, this figure is necessary as a segue to the ones that follow of the EGT model.

Specific issue Response 1: We have updated the figure to provide a much more clear segue to the following EGT model figures.

Specific issue 2: That the model can account for both slow- and fast-cycling PSA dynamics is admirable, but there is significant mismatch between the simulations in Figure 4 and the patient data. This, I believe, is partly due to PSA and tumor cell populations dropping instantaneously in the model, but dropping rather slower in patients. This could be remedied by including indicators of when patient data was measured in the plot. Stars or x's or something.

Specific issue Response 2: We have added measurement points to figure 4 to show where PSA draws were performed.

Specific issue 3: Table 1 could be better presented if Adaptive without cheaters and Adaptive with cheaters were replaced with Adaptive and the status of cheater cells was placed in the Representative Patient area. Additionally, I would suggest replacing the "---" with indefinite or some other

word because as the table is presented now, at first it seems that those therapies were not attempted rather than the excellent responses they achieved.

Specific issue Response 3: These are great suggestions and both have been changed in Table 1.

Specific issue 4: Table 1 also shows the large difference between the model predicted percent dose and the clinical data percent dose. The representative patients in Table 1 received less than 2% of the MTD, whereas the actual patients have so far received 47%. I suspect this is due to instantaneous drops in tumor cell populations in the model.

Specific issue response 4: The reviewer has correctly identified why the percentages are indeed different. A quick explanation addressing this has been added to the description of Table 1.

Specific issue 5: Lines 324 - 327 describe Table 1 and state that adaptive therapies provide significant increase in time to progression under any initial tumor condition. Only two representative patients are shown in Table 1, which are, presumably the same representative patients from line 220. This seems too few initial conditions to present to make such a sweeping claim, especially when the two representative patients sets competition coefficients are in the same category of responder.

Specific issue response 5: An analysis of the three treatment regimens for all 22 cases are included in the Supplemental Table 3. The results do indeed support the claim that adaptive therapy provides equivalent or superior results in all cases. This has been added to lines 324-327.

We have also added a quick explanation that the two representative patients actually come from two categories, the best responders and responders.

Reviewer #2 (Remarks to the Author):

Comment 1: The authors say they changed line 76 to: "efforts to translate evolutionary dynamics into a clinical setting have generally used informal, non-quantitative approaches to define the underlying Darwinian dynamics." but that change does not appear to have been made.

Response 1: We apologize. The manuscript has undergone so many revisions that it is difficult to keep up. This sentence can now be found in the first paragraph of the Discussion.

Comment 2: I found a typo: Line 124 should read "Prostate cancer, like all tumors, is an open..."

Response 2: This typo has been fixed.

REVIEWERS' COMMENTS:

Reviewer #1 (Remarks to the Author):

Apologies for the multiple rounds of revision. I feel that the current manuscript is a far stronger one than was submitted.

My concerns have been allayed.